# Effect of Thiosemicarbazone Derivatives and *Fusarium culmorum* (Wm.G. Sm.) Sacc. Infection of Winter Wheat Seedlings on Their Health Status and Soil Biological Activity

Agnieszka Jamiołkowska [1], Barbara Skwaryło-Bednarz [1,*], Elżbieta Mielniczuk [1], Franco Bisceglie [2], Giorgio Pelosi [2], Francesca Degola [2], Anna Gałązka [3] and Emilia Grzęda [3]

[1] Department of Plant Protection, University of Life Sciences in Lublin, Leszczyńskiego 7, 20-069 Lublin, Poland; aguto@wp.pl (A.J.); elzbieta.mielniczuk@up.lublin.pl (E.M.)
[2] Department of Chemistry, Life Sciences and Environmental Sustainability, University of Parma, Parco delle Scienze 11/A, 43124 Parma, Italy; franco.bisceglie@unipr.it (F.B.); giorgio.pelosi@unipr.it (G.P.); francesca.degola@unipr.it (F.D.)
[3] Department of Agriculture Microbiology, Institute of Soil Science and Plant Cultivation—State Research Institute, Czartoryskich 8, 24-100 Puławy, Poland; agalazka@iung.pulawy.pl (A.G.); egrzeda@iung.pulawy.pl (E.G.)
**\*** Correspondence: barbara.skwarylo@up.lublin.pl

**Abstract:** Climate change has an impact on agricultural production, including the greater persistence of plant pathogens in the environment. Therefore, the question arises as to how to effectively and safely protect plants by using chemicals, the number of which is decreasing each year. The aim of this study was to evaluate the fungistatic effect of 2 acetylthiophenethiosemicarbazone (2actpTS) and Cis-jasmonethiosemicarbazone (JTS) against *Fusarium culmorum* (Fc) on winter wheat seedlings. The influence of thiosemicarbazones (TSs) on the health status and phytotoxicity of seedlings and soil biological activity was investigated. Before TSs application (watering or spraying), soil was infected with *F. culmorum* (strain No. 37). The substance type and method of its application significantly influenced only the fresh weight of the seedlings. The varying phytotoxicity of the molecules depended primarily on their application method to the plants. The highest seedling phytotoxicity was recorded when compounds were applied during watering and the lowest when they were sprayed. The results showed that the application of substances 2actpTS and JTS, both in the form of watering and spraying, had a positive effect on plant health status, as expressed by the disease index (DI) value. The presence of the infectious agent and the type of chemical compound increased soil enzyme activity. The highest total number of bacteria was found in the soil that was infected with Fc and in soil samples where JTS was applied by watering and spraying. The highest utilization of amines, amides and amino acids by microorganisms was found in the samples where JTS was applied. The obtained results will be used to create intermediate products for the protection of cereals against Fusarium diseases, not only at the stage of germination and tillage of plants, but also at subsequent developmental stages.

**Keywords:** thiosemicarbazone; wheat seedlings; catalase; dehydrogenase; phytotoxicity; soil biological activity

## 1. Introduction

In the era of global warming, there is a greater persistence/survival of pathogenic fungi and an increased risk of mycotoxin plant contamination [1]. Fungi of the genus *Fusarium* and the mycotoxins produced by them (fumonisins, trichotecenes, and zeara-

lenone) pose the greatest threat to cereals [2]. *Fusarium* spp. belong to widely distributed pathogens in various climatic zones, and soil is a large reservoir of these fungi, where they survive on post-harvest plant residues. Thus, simplifications in plant cultivation, especially cereals, contribute to the accumulation of *Fusarium* spp. in the substrate, which poses a risk of infection in subsequent growing seasons [3–5]. Apart from weather conditions, factors determining cereal infection by fungi of the genus *Fusarium* include the ability to produce phytotoxic secondary metabolites, enzymes degrading cell walls and hydrolyzing fungitoxic saponins, genotype susceptibility, as well as the method and form of fertilization and forecrop type [3–8]. *Fusarium culmorum* (Fc), which infects plants at different developmental stages, is one of the particularly harmful species to cereals under various climatic conditions [9]. The risk of contamination of agricultural products with mycotoxins is not limited to raw materials. Consumption of feed polluted by livestock can lead to contamination of meat, milk, eggs and derivatives [10]. In this context, the use of synthetic fungicides is still the most effective and common method of protecting crops against pathogenic fungi. However, their application causes long-term persistence of active substances of pesticides in food and environment [11]. Scientists are looking for new solutions for safe crop protection against pathogens. One such activity is the development of new biologically active substances whose aim is to limit the growth of pathogenic fungi, while inhibiting mycotoxin production and having a low negative impact on the environment. Thiosemicarbazones (TSs) belong to such compounds. It is a class of compounds with a peculiar structure, in which the R substituent can be both aliphatic and aromatic, and a thione group coexists with a hydrazine part. The singularity of this structure leads to an extensive electron delocalization with the presence of a thione-thiol tautomerism. The key feature of TSs is the ease of their structure modification using different kinds of aldehydes or ketones. Various biological activities of TSs are being investigated in fields ranging from pharmacology to agriculture due to some highly promising properties of this class of compounds. Among them, the antimicrobial, and in particular antifungal potential is of great interest due to possible applications in plant protection [12–17]. The use of TSs in the protection of crops seems to be of particular interest. However, even though the number of antifungal studies on TSs has grown rapidly, their mechanism of action is still debated; in fact, some results highlight the ability of specific TSs to affect the cell redox balance, acting as both anti-oxidant and ROS-stimulating agents, while other studies demonstrate that certain TSs interfer with plasma membrane biogenesis and/or composition [18–21]. Recent literature attributed TS effectiveness against fungal phytopathogens belonging to the genera *Aspergillus* and *Fusarium* to the presence of complexed metal ions, indicating that the antifungal activity mainly relied on the TS ligand scaffolds [16,22,23]. This has been considered particularly interesting with respect to the overuse of metal-based pesticide formulations, which have been recklessly applied to crops for years, raising serious concerns about food safety and environmental sustainability. Research on TSs in the protection of cereals against fusariosis is very justified. The use of TS derivatives in the protection of crops requires more detailed research, not only in terms of the impact of these compounds on the health status of plants, but also to determine their mechanism of action as well as their impact on plants and soil microbiota. Therefore, the aim of this article is to evaluate the effect of two TSs molecules with anti-fungal properties, on selected biometric features of wheat seedlings and the biological activity of cultivated soil. In particular, the potential of 2-acetylthiophenethiosemicarbazone (2actpTS) and Cis-jasmonethiosemicarbazone (JTS), two TSs previously characterized that were obtained from natural scaffolds (2-acetylthiophene and jasmonic acid respectively), was investigated against the phytopathogen *Fusarium culmorum* directly on winter wheat, taking into consideration different phytosanitary aspects. The results of the conducted research will be fundamental in deciding on further work addressed to the development of commercial preparations and their formulation, in order to effectively protect cereals against fusarioses.

## 2. Materials and Methods

### 2.1. Pant Material

The experiment was carried out in a growth chamber of the Department of Plant Protection of the University of Life Sciences in Lublin (Poland). Winter wheat seedlings (*Triticum aestivum* L.), cultivar 'Hondia' from DANKO Hodowla Roślin Sp. z o.o., Choryń, Kościan, Poland, were used in the experiments.

### 2.2. Fusarium Culmorum Inoculum

*Fusarium culmorum* (Wm.G. Sm.) Sacc. was used in the study (Fc strain No. 37), as a representative isolate from winter wheat kernels. The *F. culmorum* strain was derived from the own collection of the Department of Plant Protection/Department of Phytopathology and Mycology, University of Life Sciences in Lublin (Poland). In accordance with the methodology described by Ossowicki et al. [24], the inoculum of the tested fungus strain was prepared in the form of 14-day fungus culture grown on potato dextrose agar (PDA, Difco, Becton, Dickinson and Company Sparks, Sparks, Maryland, USA; 39 g/1000 mL) in Petri dishes. After fungus culture incubation at 20 °C, the plant growth soil was inoculated with the fungus using one 6 mm fragment per 10 mL of the soil. In the control combination (C0), the soil was mixed with PDA fragments.

### 2.3. Chemical Synthesis of Thiosemicarbazone Derivatives

The compounds under study, 2actpTS (2-acetylthiophenethiosemicarbazone) and JTS (Cis-jasmonethiosemicarbazone), were synthesized and characterized as previously described (Figure 1) [20,23]. All reagents for the syntheses were purchased form Sigma Aldrich. The 1H-NMR spectra were recorded using a Bruker Anova spectrometer (Bruker, Billerica, MA, USA) at 400 MHz. ESI-MS analysis was performed using a Waters Acquity Ultra Performance ESI-MS spectrometer (Waters Corp., Milford, MA, USA) with a single quadrupole detector. Elemental analyses were performed on a CHNS ThermoFischer (Rodano, MI, Italy). Compounds were dissolved in DMSO.

**Figure 1.** Schematic representation of the structures of the two thiosemicarbazone derivatives used in this study (2actpTS and JTS).

### 2.4. Thiosemicarbazones Treatments

Winter wheat seeds were surface-disinfected with 0.1% sodium hypochlorite for 1 min, and subsequently rinsed three times in distilled water [25]. The seeds were germinated on glass plates filled with sterile filter paper. When well-formed sprouts emerged, the seedlings were planted in 1 L plastic pots (seven days after soil infection) and filled with soil from wheat cultivation with the grain size composition of medium clay (fraction content in %: 40—sand, 20—silt, 40—clay) with the addition of sand in a ratio of 4:1 (pH 6.5) infected with *F. culmorum* (Fc). Twenty-five germinated wheat seedlings were planted in each pot. After five days of plant growth in the infected soil, the seedlings

were treated once with the test substances according to the following scheme: (1) fungus-infected plants (planted in fungus-contaminated soil) (C1), (2) fungus-infected plants watered with 2actpTS (2actpTS/W+Fc), (3) fungus-infected plants sprayed with 2actpTS (2actpTS/S+Fc), (4) non-infected plants watered with 2actpTS (2actpTS/W), (5) non-infected plants sprayed with 2actpTS (2actpTS/S), (6) fungus-infected plants watered with JTS (JTS/W+Fc), (7) fungus infected plants sprayed with JTS (JTS/S+Fc), (8) non-infected plants watered with JTS (JTS/W), and (9) non-infected plants sprayed with JTS (JTS/S). The absolute control (C0) consisted of wheat seedlings planted in a non-infected soil (without fungus) and not treated with the test substances. Substances 2actpTS and JTS were used in the experiment at a concentration of 100 ppm. For plant watering and spraying, 25 mL of substance/pot was applied. An adjuvant (Tween80®) was added to the spraying formula for a better substance adherence to leaf surface. Three replicates were prepared for each experimental combination. Plants were placed in a growth chamber at $22 \pm 1$ °C and 85% relative air humidity with a 14-h photoperiod and watered with sterile water as needed.

### 2.5. Molecule Phytotoxicity (PI) and Plant Disease Index (DI) Determination

The phytotoxicity of molecules (phytotoxicity index—PI) was assessed two weeks after their application to plants using a 6-point scale: 0—no leaf damage; 1—slight damage to the leaf blade covering less than 10% of leaf area; 2—damage to the leaf blade covering 10–25% of leaf area; 3—damage to the leaf blade covering 26–50% of leaf area; 4—damage to the leaf blade covering 51–75% of leaf area; 5—damage to the leaf blade covering 76–100% of leaf area.

After four weeks of the experiment, the fungistatic effect of the molecules was assessed by determining the degree of seedling infection with the pathogenic agent (*F. culmorum*), i.e., plant disease index (DI) using a 5-point scale: 0—no symptoms, 5—necrosis covering 76–100% of leaf sheath surface and seedling roots [26]. The PI and DI were estimated for each experimental combination (i.e., replication) using the Townsend-Heuberger formula [27]: disease index (PI/DI) = $(\sum a/\sum b) \times 100$, where a = sum of the products of the numerical scale index (degree of infection) and the corresponding number of plants, and b = total number of plants tested multiplied by the largest index of the numerical scale. Subsequently, the mean phytotoxicity index and mean disease index for each combination were calculated. In order to meet Koch's postulates, a mycological analysis of plants from individual combinations of the experiment was performed. Fifty root and leaf sheath fragments of wheat seedlings from each experimental combination were analyzed. Mycological analysis of the infected wheat seedlings identified the test *F. culmorum* strain as the cause of seedling blight.

### 2.6. Assessment of Seedling Fresh and Dry Weight

The analysis was carried out four weeks after the application of the molecules to the plants. From each experimental combination, 30 seedlings (10 plants from each replicate/pot) were collected. The seedlings were cleaned of soil debris and weighed. The measurement results were expressed in g $fw^{-1}$. After drying the plants (in a ventilated room at a temperature of 23–25 °C for five days), the test material was weighed and the obtained results were expressed in g $fw^{-1}$.

### 2.7. Determination of Catalase and Dehydrogenase Activity in Soil

Catalase activity was determined by the method of Johnson and Temple [28], which consisted in incubating the soil with added hydrogen peroxide (natural enzyme substrate). The results were given in units of catalase activity, equal to mg $H_2O_2$ $g^{-1}$ dw $min^{-1}$. Soil dehydrogenase activity was determined according to the methodology described by Casida [29]. The method is based on the action of 2,3,5-triphenyltetrazole chloride (TTC) as an artificial hydrogen and electron acceptor. The enzyme activity was measured spec-

trophotometrically at 485 nm. Determinations of dehydrogenase activity for each sample were carried out in triplicate and the results were expressed as arithmetic means, equal to µg TPF $g^{-1}$ dw $d^{-1}$.

### 2.8. Analysis of Microorganism Abundance in Soil

Soil was collected for testing two weeks after application of the molecules to the plants, in accordance with the relevant methodology [30,31]. The soil sample was averaged for each experimental combination and then 10 g of soil was weighed for further analyses (three replicates for each experimental combination). Soil solutions were prepared from individual samples at dilutions of $10^{-1}$ to $10^{-7}$. The total bacterial count was determined on nutrient agar (Difco; Becton, Dickinson and Company Sparks, Sparks, Maryland, USA) using $10^{-5}$, $10^{-6}$ and $10^{-7}$ solutions. Tryptic soy agar (Difco; Becton, Dickinson and Company Sparks, USA) in $10^{-4}$, $10^{-5}$ and $10^{-6}$ dilutions were used, while Pseudomonas agar F (Difco; Becton, Dickinson and Company Sparks, USA) in $10^{-2}$, $10^{-3}$ and $10^{-4}$ dilutions were used for Pseudomonas spp. To isolate Bacillus spp., soil dilutions were heated for 20 min at 80 °C [30]. The total number of fungi in each soil sample was determined using Martin's medium at dilutions of $10^{-2}$, $10^{-3}$ and $10^{-4}$. Petri dishes were stored in the dark at 24 °C for two to seven days. After incubation, the number of microorganisms was converted to CFU $g^{-1}$ soil dw (colony-forming units/g soil dry weight).

### 2.9. Analysis of Soil Functional Biodiversity Using Biolog EcoPlates

The functional biodiversity of microorganisms was studied using Biolog EcoPlatesTM (Biolog, Inc., Hayward, CA, USA). One gram of soil was added to 99 mL of sterile water and shaken for 20 min at room temperature. The suspension was left for 30 min at 4 °C for the particles to settle to the bottom of the vessel [32]. Each well was filled with 120 µL of soil suspension and incubated at 25 °C for seven days in an OmniLog® ID System multiplicate reader (Biolog, Inc., Hayward, CA, USA). Plates were prepared in triplicate for each experimental sample. Very intense metabolic activity of 31 carbon sources was observed on the plates after 96 h of incubation. The results of this analysis were confirmed every 24 h using the Micro Station ID system ($\lambda = 590$ nm) up to 144 h of incubation (Biologist, Inc., Howard, CA, USA). Microbial activity was confirmed in five groups of compounds (amines and amides, amino acids, carbohydrates, carboxylic acid and polymers). The cluster analyses and PCA were performed on standardized data from the average absorbance values at 96 h (Biolog EcoPlates). This is a common analysis recommended by other authors also. Average well color development (AWCD) was calculated according to the equation: AWCD = $[\Sigma (C - R)]/n$ where C represents the absorbance value of control wells (mean of three controls), and R is the mean absorbance of the response wells (three wells per carbon substrate).

### 2.10. Statistical Analysis

The results from the obtained data were statistically analyzed using the Statistica.pl package (10) (Stat. Soft. Inc., Tulsa, OK, USA). The collected data were subjected to the one-way analysis of variance (ANOVA) for comparison of the means. Significant differences were calculated using Tukey's post hoc test at a significance level of $p < 0.05$. Cluster analyses were performed on data standardized from the average absorbance values 96 h post inoculum (Biolog EcoPlates). The results were also subjected to principal component analysis (PCA) [33].

## 3. Results

### 3.1. Phytotoxicity of Compounds and Plant Infestation with F. culmorum

Molecules 2actpTS and JTS applied at a concentration of 100 ppm showed different phytotoxicity to wheat seedlings. Whitening of leaf tips and their twisting were observed

in all experimental combinations (Figure 2). The phytotoxicity of the molecules depended primarily on the method of their application.

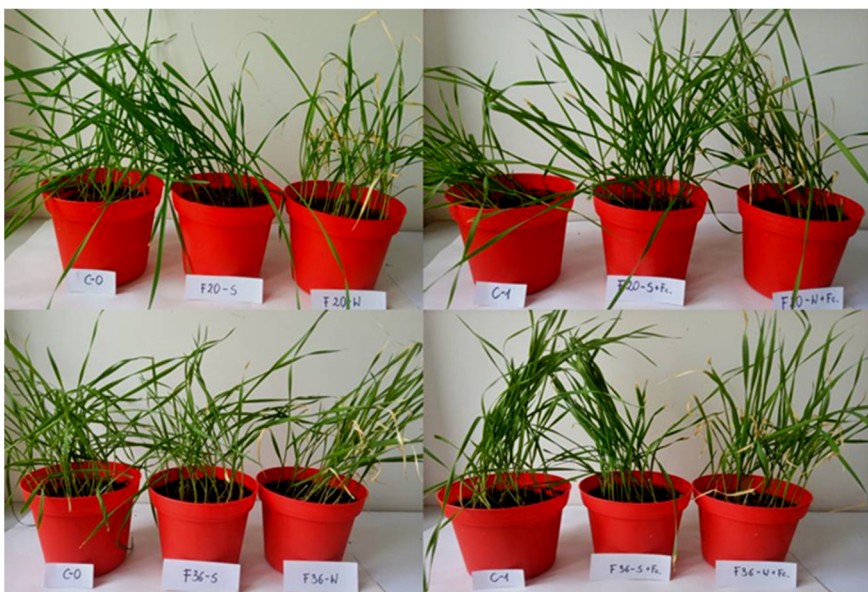

**Figure 2.** Phytotoxicity symptoms in wheat seedlings after 2actpTS and JTS application.

The highest phytotoxicity was recorded when seedlings were watered with JTS (JTS/W—53.00) and 2actpTS (2actpTS/W—46.00) molecules, and the lowest for plants sprayed with 2actpTS (2actpTS/S—5.00) (Table 1). This also indicated a more efficient uptake of molecules by seedlings, and thus a higher effectiveness of the compounds. The phytotoxicity of the molecules applied to Fc-infected seedlings in the form of watering (2actpTS/W+Fc—44.00, JTS/W+Fc—5.00) was significantly lower than the above combination (Table 1). It can be assumed that Fc present in the infected soil contributed to the neutralization of molecule toxicity, and thus the reduction of the phytotoxicity effect.

The observations showed that plants with disease symptoms were observed in all combinations of the experiment with wheat kernels infected by *F. culmorum* (Fc). Infected plants were characterized by necrosis of the roots and sub-cotyledon parts, root system reduction and seedling growth inhibition. The results demonstrated that the application of substances 2actpTS and JTS, both in the form of watering and spraying, exerted a positive effect on plant health expressed in the value of the disease index (DI). Statistically, the lowest value of the disease index was recorded for seedlings growing in the soil without Fc inoculation, with the application of 2actpTS in the form of a watering (2actpTS/W—11.00) (Table 1). This value did not differ significantly from the disease indices in the control combination (C0—14.30) and in the combination where wheat seedlings were sprayed with JTS (JTS/S—12.00), as well as in combinations 2actpTS in the form of spraying (2actpTS/S—13.30). Statistically the highest value of the disease index was recorded in the experimental combinations with Fc soil inoculation, especially in combination where plants were spraying with JTS and inoculated with Fc (JTS/S+Fc—23.70) (Table 1). The form of application of molecules influenced the disease index of infected seedlings. Plants infected with Fc and then watered had a lower DI than the plants infected and sprayed with molecules. The isolates of the studied *Fusarium* species accounted for 78.5% of the total isolations from wheat seedlings.

**Table 1.** Phytotoxicity, plant biomass, and disease index of wheat (*Triticum aestivum* L.) seedlings after 2actpTS and JTS application.

| Experimental Combination | Mean Phytotoxicity Index (PI) | Fresh Weight (g fw$^{-1}$) | Dry Weight (g dw$^{-1}$) | Mean Disease Index (DI) |
|---|---|---|---|---|
| 2actpTS/W | 46.00 [a] | 3.192 [abcd] | 0.471 [b] | 11.00 [a] |
| 2actpTS/S | 5.00 [c] | 4.498 [d] | 0.638 [ab] | 13.30 [ab] |
| 2actpTS/W+Fc | 44.00 [a] | 4.147 [bcd] | 0.635 [ab] | 15.70 [bc] |
| 2actpTS/S+Fc | 12.00 [bc] | 5.903 [a] | 0.705 [a] | 16.70 [c] |
| JTS/W | 53.00 [a] | 4.046 [cd] | 0.746 [a] | 15.00 [bc] |
| JTS/S | 13.00 [bc] | 4.407 [abcd] | 0.592 [ab] | 12.00 [ab] |
| JTS/W+Fc | 5.00 [c] | 3.732 [cd] | 0.572 [ab] | 14.30 [ab] |
| JTS/S+Fc | 19.00 [b] | 5.02 [abc] | 0.667 [a] | 23.70 [d] |
| C1 | 0.00 [c] | 5.644 [ab] | 0.654 [ab] | 17.70 [c] |
| C0 | 0.00 [c] | 4.934 [abc] | 0.614 [ab] | 14.30 [ab] |

Treatment means separated by different letters are significantly different (Tukey's mean separation test, $p < 0.05$); (n = 3).

### 3.2. Plant Biomass Production

The plant biomass was expressed as fresh and dry weight of seedlings. The results of fresh and dry weight measurements of plants are presented in Table 1. The type of substance and method of its application significantly influenced the decrease in the fresh weight of wheat seedlings. The lowest fresh weight had seedlings watered with 2actpTS and JTS molecules (2actpTS/W—3.192 g fw$^{-1}$; JTS/W—4.046 g fw$^{-1}$), and seedlings infected with fungus and watered with molecules (2actpTS/W+Fc—4.147 g fw$^{-1}$, JTS/W+Fc—3.732 g fw$^{-1}$). The above values significantly differed from the controls (C1—5.644 g fw$^{-1}$, C0—4.935 g fw$^{-1}$) (Table 1). Fresh weight of seedlings infected with the fungus and sprayed with the molecules was higher (2actpTS/S+Fc—5.903 g fw$^{-1}$, JTS/S+Fc—5.02 g fw$^{-1}$) than the fresh weight of seedlings only subjected to spraying (2actpTS/S—4.498 g fw$^{-1}$, JTS/S—4.407 g fw$^{-1}$), but these values were not significantly different from each other or the control (Table 1). Wheat seedlings watered with 2actpTS (2actpTS/W—0.471 g fw$^{-1}$) had the lowest dry weight, and those watered with JTS (JTS/W—0.746 g fw$^{-1}$) had the highest. The dry weight of seedlings infected with the fungus and sprayed was higher (2actpTS/S+Fc—0.705 g fw$^{-1}$, JTS/S+Fc—0.667 g fw$^{-1}$) than the dry weight of Fc-infected, molecule-watered seedlings (2actpTS/W+Fc—0.635 g fw$^{-1}$, JTS/W+Fc—0.572 g fw$^{-1}$), but these values did not significantly differ from each other or the control (Table 1). The type of substance and method of its application did not significantly influence the dry weight of wheat seedlings.

### 3.3. Biological Activity of the Soil

The obtained results demonstrated that the presence of the Fc infectious agent, type of chemical substance and the form of its application influenced the activity of soil enzymes. The highest catalase activity was recorded in the Fc-infected soil and when 2actpTS and JTS were applied in the form of watering (C1—0.092 mg $H_2O_2$ g$^{-1}$ dw min$^{-1}$, 2actpTS/W+Fc—0.088 mg $H_2O_2$ g$^{-1}$ dw min$^{-1}$, 2actpTS/W—0.081 mg $H_2O_2$ g$^{-1}$ dw min$^{-1}$, JTS/W+Fc—0.078 mg $H_2O_2$ g$^{-1}$ dw min$^{-1}$). Statistically lower catalase activity was recorded for the absolute control (C0—0.058 mg $H_2O_2$ g$^{-1}$ dw min$^{-1}$) and for the soil where 2actpTS and JTS were applied as a spray (2actpTS/S—0.050 mg $H_2O_2$ g$^{-1}$ dw min$^{-1}$, JTS/S—0.040 mg $H_2O_2$ g$^{-1}$ dw min$^{-1}$) (Table 2).

Dehydrogenase activity depended on the presence of the infectious agent (Fc) in the soil. The highest dehydrogenase activity, statistically different from other experimental combinations, was found in the control soil infected with Fc (C1—61.67 µg TPF g$^{-1}$ dw d$^{-1}$). High dehydrogenase activity was also recorded in soil collected from seedlings in-

fected with Fc and watered and sprayed with molecules (JTS/W+Fc—54.37 µg TPF g$^{-1}$ dw d$^{-1}$, 2actpTS/S+Fc—50.28 µg TPF g$^{-1}$ dw d$^{-1}$, JTS/S+Fc—47.93 µg TPF g$^{-1}$ dw d$^{-1}$). Significantly lower enzyme activity was recorded in soil not infected with fungi, where 2actpTS and JTS were applied in various forms (2actpTS/S—46.34 µg TPF g$^{-1}$ dw d$^{-1}$, JTS/W—44.21 µg TPF g$^{-1}$ dw d$^{-1}$), and its value did not differ significantly from the absolute control (C0). The exception was the 2actpTS/W combination, where the enzyme activity was significantly higher than in other soil samples without the infectious agent (Table 2).

**Table 2.** Biological activity of the soil with wheat seedlings (*Triticum aestivum* L.) after application of 2actpTS and JTS molecules.

| Experimental Combination | Total Number of Bacteria (10$^7$ CFU g$^{-1}$ dw Soil) | Total Number Of Fungi (10$^4$ CFU G$^{-1}$ Dw Soil) | Catalase Activity (mg H$_2$O$_2$ g$^{-1}$ dw min$^{-1}$) | DEHYDROGENASE Activity (µg TPF g$^{-1}$ dw d$^{-1}$) |
|---|---|---|---|---|
| 2actpTS/W | 114.868 [b] | 53.749 [b] | 0.081 [b] | 54.55 [b] |
| 2actpTS/S | 64.542 [c] | 64.542 [b] | 0.050 [e] | 46.34 [c] |
| 2actpTS/W+Fc | 151.254 [b] | 76.449 [a] | 0.088 [a] | 39.58 [d] |
| 2actpTS/S+Fc | 81.917 [c] | 41.830 [b] | 0.053 [de] | 50.28 [bc] |
| JTS/W | 289.116 [a] | 42.517 [b] | 0.065 [c] | 44.21 [cd] |
| JTS/S | 233.706 [a] | 49.348 [b] | 0.040 [f] | 42.26 [d] |
| JTS/W+Fc | 45.947 [c] | 55.483 [b] | 0.078 [b] | 54.37 [b] |
| JTS/S+Fc | 75.615 [c] | 49.217 [b] | 0.048 [e] | 47.93 [bc] |
| C1 | 285.277 [a] | 79.948 [a] | 0.092 [a] | 61.67 [a] |
| C0 | 141.304 [b] | 65.620 [b] | 0.058 [d] | 47.17 [c] |

Treatment means separated by different letters are significantly different (Tukey's mean separation test, *p* < 0.05); (n = 3).

A significant increase in the total count of bacteria was found in Fc-infected control soil (C1) and in soil samples where JTS was applied by watering and spraying (JTS/S, JTS/W) (Table 2). Moreover, a significant increase in the total number of fungi was found in the C1 and F20/W+Fc combinations compared to control (Table 2).

*3.4. Biolog EcoPlates Analysis*

Carbohydrates and carboxylic acids were the most actively utilized among the five groups of compounds tested on Biolog EcoPlates plates (Table 3); while amino acids were the least utilized compounds. There were slight differences in the utilization of particular compound groups in a given combination compared to control. Heatmap was calculated based on the data obtained after a 96-h incubation of EcoPlates (Figure 3). Of 31 compounds, those from samples JTS/S and JTS/W+Fc were most actively utilized by microorganisms. The lowest metabolic activity of microorganisms based on the utilization of 31 compounds was observed in soil C0, JTS/S+Fc and 2actpTS/S+Fc.

Ward's clustering was performed for 31 compounds after a 96-h incubation of EcoPlates, calculating dehydrogenase activity and total number of bacteria and fungi. The results of clustering analysis for all samples are shown in Figure 4. The analysis was carried out using all measured microbiological properties of the soil samples. The first group of the most similar samples in terms of soil microbiological properties included the JTS/S+Fc, JTS/W and 2actpTS/S+Fc. The second group consisted of the 2actpTS/W, JTS/W+Fc, JTS/S, 2actpTS/S, 2actpTS/W+Fc and C1 samples. The C0 control soil formed a separate, third group.

**Table 3.** Utilization percentage (%) of substrate groups in soil samples.

| Experimental Combination | Amines and Amides | Amino Acids | Carbohydrates | Carboxylic Acids | Polymers |
|---|---|---|---|---|---|
| 2actpTS/W | 6.888 [b] | 19.752 [a] | 29.556 [a] | 29.909 [b] | 13.894 [a] |
| 2actpTS/S | 6.824 [b] | 18.112 [b] | 29.228 [a] | 31.171 [a] | 14.665 [a] |
| 2actpTS/W+Fc | 6.150 [c] | 19.848 [a] | 28.621 [b] | 30.543 [b] | 14.838 [a] |
| 2actpTS/S+Fc | 6.528 [c] | 20.443 [a] | 29.372 [a] | 29.082 [b] | 14.576 [a] |
| JTS/W | 7.461 [a] | 19.232 [b] | 27.344 [b] | 31.570 [a] | 14.392 [a] |
| JTS/S | 7.238 [a] | 18.781 [b] | 29.635 [a] | 29.874 [b] | 14.472 [a] |
| JTS/W+Fc | 7.025 [b] | 19.287 [b] | 28.481 [b] | 30.650 [b] | 14.557 [a] |
| JTS/S+Fc | 8.101 [a] | 19.556 [a] | 29.292 [a] | 29.132 [b] | 13.920 [a] |
| C1 | 6.444 [c] | 19.294 [b] | 30.226 [a] | 29.840 [b] | 14.196 [a] |
| C0 | 6.043 [c] | 19.093 [b] | 28.485 [b] | 31.858 [a] | 14.521 [a] |

Treatment means separated by different letters are significantly different (Tukey's mean separation test, $p < 0.05$); (n = 3).

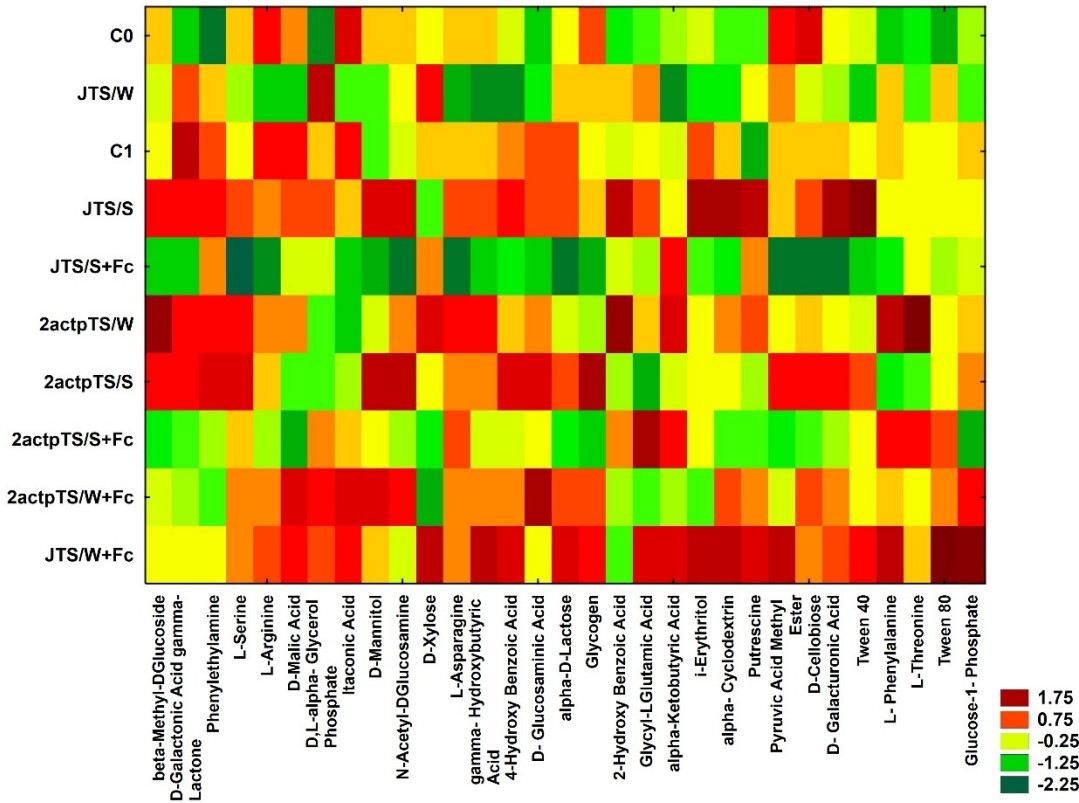

**Figure 3.** Heatmap based on the analysis of 31 carbon sources after a 96-h incubation of Biolog EcoPlates. Plates were replicated in triplicate.

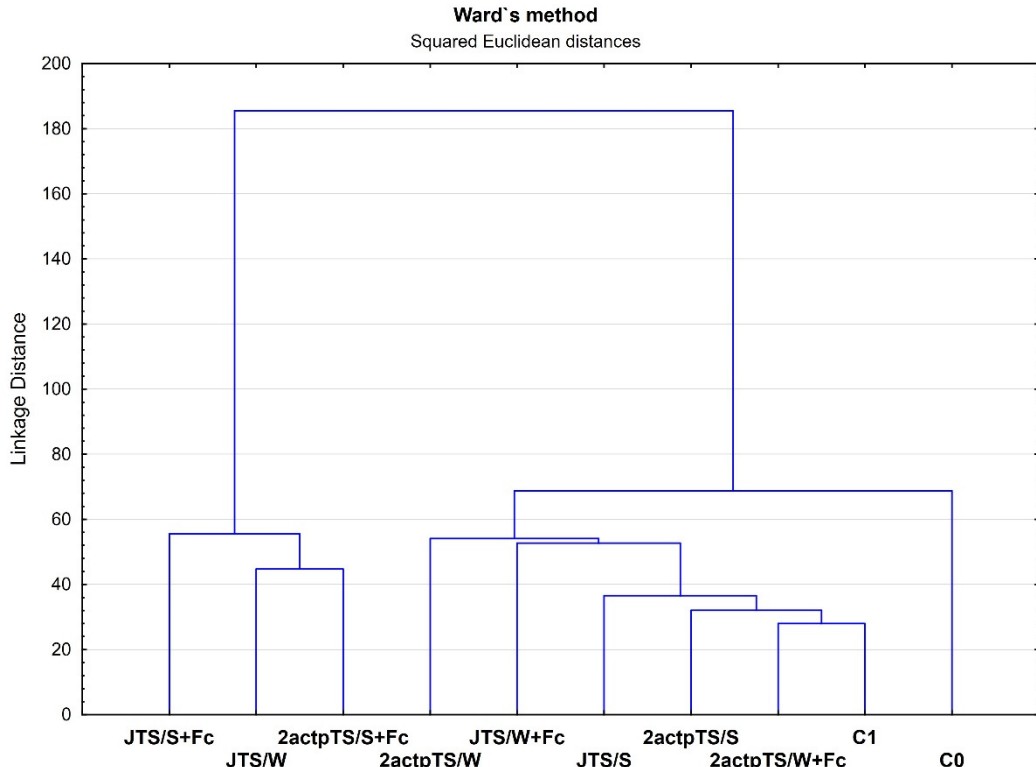

**Figure 4.** Hierarchical dendrogram of samples using Ward's clustering method with bond distances. Boundaries of Sneath's criteria, restrictive (33%) and less restrictive (66%), are marked on the dendrogram.

Table 4 shows the biodiversity indicators calculated on the basis of data obtained from 96h incubation of Biolog EcoPlates. No statistically significant differences were found between the samples in Shannon's general diversity index (H') and substrate evenness (E). Small differences were found in substrate richness (R) and average well-color development (AWCD590).

**Table 4.** Changes in microorganisms metabolic diversity in soils as evaluated by: Shannon's general diversity index ($H'$), substrate richness (R), substrate evenness (E) and average well-color development ($AWCD_{590}$) obtained in the Biolog EcoPlates incubated for 96h.

| Experimental Combination | H' | R | E | AWCD590 |
|---|---|---|---|---|
| 2actpTS/W | 3.405 [a] | 30.333 [a] | 0.992 [a] | 2.101 [a] |
| 2actpTS/S | 3.393 [a] | 29.667 [a] | 0.994 [a] | 2.016 [a] |
| 2actpTS/W+Fc | 3.395 [a] | 30.333 [a] | 0.992 [a] | 2.096 [a] |
| 2actpTS/S+Fc | 3.395 [a] | 30.000 [a] | 0.995 [a] | 1.896 [b] |
| JTS/W | 3.390 [a] | 29.333 [a] | 0.994 [a] | 1.918 [a] |
| JTS/S | 3.412 [a] | 31.000 [a] | 0.993 [a] | 2.201 [a] |
| JTS/W+Fc | 3.396 [a] | 30.000 [a] | 0.995 [a] | 2.122 a |
| JTS/S+Fc | 3.399 [a] | 30.333 [a] | 0.990 [a] | 1.795 [b] |
| C1 | 3.398 [a] | 30.000 [a] | 0.996 [a] | 2.010 [a] |
| C0 | 3.362 [a] | 28.667 [b] | 0.998 [a] | 1.916 [a] |

Treatment means separated by different letters are significantly different (Tukey's mean separation test, $p < 0.05$); (n = 3).

PCA was used to assess the correlation between all samples and soil microbial activity for different soils. To reduce dataset dimensionality, PCA was used to compare the relationship between enzyme activity, total bacterial and fungal counts and the metabolic

profile of microorganisms obtained after a 96h incubation of EcoPlates. PCA values indicated that the first two major components (PC) accounted for 34.50% and 20.18% of the total sample variance. The PCA plot of the first two PCs showed that all samples varied with distance. All microbiological properties were positively associated with samples C1, 2actpTS/W, JTS/W+Fc, JTS/S, 2actpTS/S and 2actpTS/W+Fc (Figure 5).

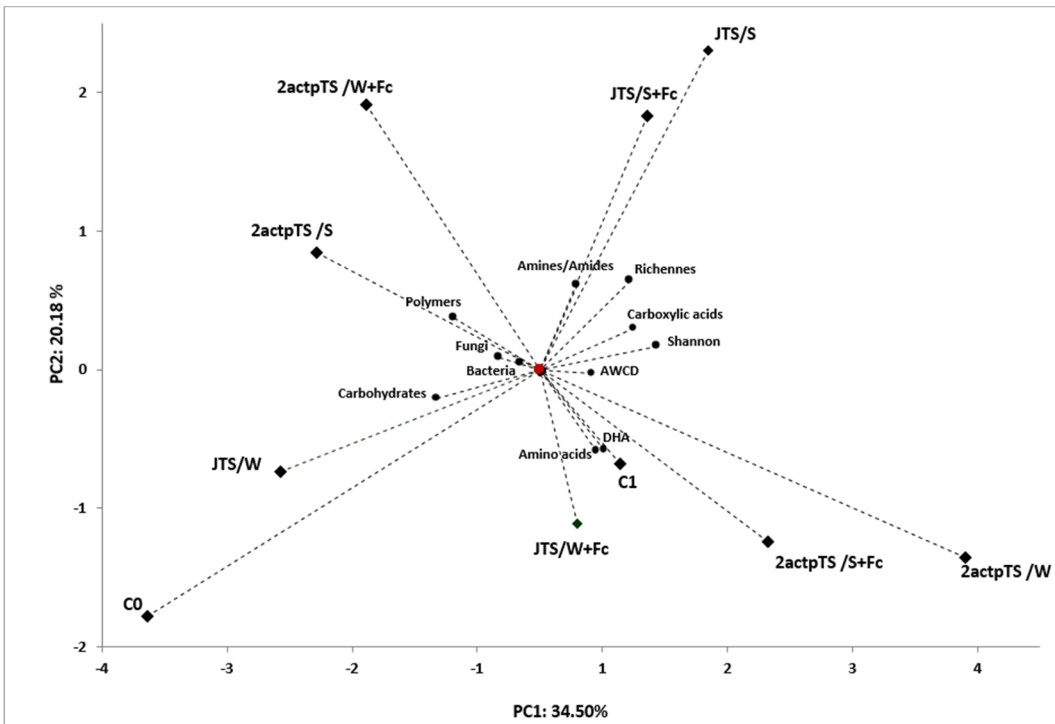

**Figure 5.** Hierarchical dendrogram of samples using Ward's clustering method with bond distances. Boundaries of Sneath's criteria, restrictive (33%) and less restrictive (66%), are marked on the dendrogram.

## 4. Discussion

The use of natural products as scaffolds for new generation pesticides is regarded as a privileged starting point to obtain selective and effective antifungals that are less polluting for the environment, as they are, in most cases, secondary metabolites produced by organisms to protect themselves from both biotic of abiotic stresses. With this purpose, the addition of thiosemicarbazide to various natural compound derivatives was recently used, and successfully produced new molecules with promising antifungal and anti-mycotoxigenic potential against some phytopathogenic species [14,22,23,34,35]. Jasmonic acid derivatives in particular showed a containment effect on both fungal growth/development and secondary metabolism of phytopathogenic species, being thus extremely interesting for plant health control strategies and food security purposes. In fact, Jasmonic acid (JA) is considered to play an important role in plant defense reactions. It is also known that JA, along with other phytohormones (e.g., salicylic acid, ethylene, indole-3-acetic acid, abscisic acid, cytokinin and gibberellin), is involved in triggering the host defense response to biotic (such as pathogen infection or herbivore attack) and abiotic environmental stresses [36,37]. As with other plant hormones, it can also influence fungal physiology; it is worth noting that JA may affect the growth of mycelium and production of conidia and their germination and sexual reproduction, which suggests that the molecules may interfere with specific signal transduction processes in fungi [38]. Therefore, two thiosemicarbazone derivatives that previously proved to be effective in vitro against some fungal phytopathogens affecting cereals were tested in laboratory conditions against *Fusarium culmorum*, in order to evaluate their fungistatic potential when directly applied on winter wheat seedlings. The influence of the molecules on se-

lected plant parameters (phytotoxicity, health status and biomass production), as well as biological activity of the wheat soil mycobiota, was assessed in relation to the method of administration. Previous studies, that showed interesting in vitro antifungal activity of JTS against the phytopathogenic species *Aspergillus flavus*, *Fusarium sporotrichioides*, *F. culmorum*, *F. graminearum* and *F. poae*, [22,23] were confirmed by our results, that proved the fungistatic effect of thiosemicarbazone derivatives against the pathogenic fungus *Fusarium culmorum* also *in vivo*, showing a particularly effective action of this compound in reducing the infestation of wheat seedlings by Fusarium Seedling Blight Disease.

From the perspective of the application of thiosemicarbazones (2-acetylthiophenethiosemicarbazone and Cis-jasmonethiosemicarbazone) in plant protection, their direct impact on plants and soil biological activity also seems to be of interest. The conducted research demonstrated that the effect of molecule interaction depended on the type of compound and method of its application. The tested molecules, apart from having a direct fungistatic effect on *F. culmorum*, showed potent effects on plants. The substance JTS was more phytotoxic than 2actpTS, just as the application of compounds by watering was more harmful to plants than application by spraying. The results of this study are highly important due to the serious threat posed by cereal Fusarium diseases, where protection of cereals against soil *Fusarium* spp., already at the stage of seed germination and tillage of seedlings, is particularly important [4]. These results can be used to create an intermediate product for cereal protection against Fusarium diseases. These studies, however, need further continuation in order to determine the appropriate dose and formulation of the preparation to be able to reduce the observed phytotoxicity effect.

The tested substances decreased the fresh weight of seedlings. It can be assumed that the molecules, due to the increased phytotoxicity, limited plant nutrient absorption from soil, negatively affecting fresh weight production. Positive results have been obtained for substance application in the form of spraying. It can be assumed that the decrease in plant fresh weight could be additionally caused by the presence of *F. culmorum* in the soil. The conducted research showed a significant influence of the molecules on the soil microbiota and its biological activity. Changes in the activity of soil microorganisms are an indicator of the influence of chemical/synthetic compounds on the environment [39]. Enzymatic activity is one of the parameters measured in the assessment of soil quality [40–42]. Dehydrogenases, phosphatases, urease, proteases, arylsulfatase, invertase and amidase are the most commonly studied soil enzymes [43–45]. The activity of dehydrogenases and catalase in soil is an indicator of the intensity of respiratory metabolism of all soil microbial populations, which is used to determine the total soil microbial activity [46]. Catalase (CA) is an intracellular enzyme found in aerobic and anaerobic microorganisms, involved in the breakdown of toxic $H_2O_2$ [47,48]. Catalase activity in soil is associated with soil microorganisms [49] and additionally depends on the content of organic matter, biomass, $O_2$ uptake, $CO_2$ release, as well as the activity of dehydrogenases, glucosidase amidase and phosphodiesterase [50]. The activity of dehydrogenases is commonly used to assess factors that have a negative effect on soil microorganisms [51], and is an indicator of the level of soil contamination [52,53]. Many studies have indicated varying effects of toxic substances (including pesticides) on dehydrogenase activity in soil; from negative (propiconazole, profenophos, fenophos, pretilachlor, chlorpyrifos, teflubenzuron) through neutral (fenamiphos) to stimulating effects (endosulfan) [45,54–62]. Medo et al. [55] reported that herbicides applied in the recommended doses (Dimethachlor—1.0 kg ha$^{-1}$, Linuron—0.8 kg ha$^{-1}$) had no effect on the transient increase of soil microbial biomass, changes in the microbial community and dehydrogenase activity. However, in 100-fold higher doses, these herbicides contributed to an increase in the tested parameters, decrease in soil dehydrogenase activity and changes in microbial communities. The current study showed that the presence of an infectious agent in the soil (*F. culmorum*) and watering seedlings with thiosemicarbazones significantly increased the activity of catalase and dehydrogenase in soil [55]. The research showed that

the activity of dehydrogenases in the soil under spring barley was slightly affected by the type of fungicides with a different mechanism of action (Swing Top 183 SC and Unix 75 WG) [63]. Factors, such as the fungicide dose, method of soil utilization and fungicide action time, exerted a considerable impact, and high doses of fungicides, causing soil contamination, significantly inhibited dehydrogenase activity [63]. Soil enzymatic activity, including dehydrogenase activity, depends mainly on the mass of microorganisms living in soil and available carbon sources [64]. It can be assumed that the introduction of excessive amounts of chemical substances (molecules) or heavy metals may eliminate species or genera of microorganisms that are not very resistant to these substances. This has been confirmed by the studies of other authors [56,65–67]. It is worth emphasizing that natural plant products, such as azadirachtin (natural pesticide), used in plant protection have a promoting effect on the activity of dehydrogenase and catalase in soil [68]. An important factor determining the development of plants are the communities of soil microorganisms and their biological activity [31]. Measurements of physiological profiles at the level of soil microbial communities can be used as an indication of biological activity and natural biochemical processes in soil [69]. In the present study, a significant relationship was observed between the total number of bacteria and fungi, their enzymatic activity and metabolic profile. It was shown that the increase in the total number of bacteria occurred during severe biotic and abiotic stress (presence of an infectious agent in soil and toxic substances/molecules). The experiments demonstrated that Cis-jasmonethiosemicarbazone (JTS) was more toxic to soil microorganisms (including the infectious factor—Fc) and plants than 2-acetylthiophenethiosemicarbazone (2actpTS). The number of different groups of microorganisms is a marker of changes occurring in the soil [66]. It should be noted that there are microorganisms in the soil environment, including bacteria, which in the process of natural bioremediation (bioattenuation) remove, reduce, degrade or immobilize pollutants, thereby restoring contaminated sites to a relatively pure, non-toxic environment [70]. Neutralization of toxic compounds (e.g., xenobiotics) by microorganisms may occur through biodegradation (oxidation and degradation), assimilation or biotransformation (transformation into non-toxic chemical compounds) [71,72].

## 5. Conclusions

The concept of effective control of cereal Fusarium diseases through the use of thiosemicarbazone derivatives is a very interesting alternative in crop protection. It consists in selecting the appropriate dose/concentration of fungistatic thiosemicarbazones, especially to control Fusarium Seedling Blight Disease in wheat. Thiosemicarbazones reduce mycelial growth/development of *Fusarium* spp. and affect their metabolism by inhibiting mycotoxin production [22]. The obtained results can be used to create an intermediate product for the protection of cereals against Fusarium diseases, not only at the stage of germination and tillage of plants, but also at subsequent developmental stages. The beneficial effect of thiosemicarbazones on soil biological activity, indirectly contributes to reducing the abundance of soil pathogens. An increase in the overall abundance of bacteria in the soil also leads to faster neutralization of toxic substances in the soil. However, these studies need to be continued to determine the appropriate dose and form of preparation application to reduce the observed phytotoxicity effect. This creates further challenges for both the agrochemical industry and phytopharmaceutical companies to commercialize novel approaches, i.e., how to effectively market new products and develop effective disease control strategies without reducing plant productivity. Integrating fungicide application with other non-chemical methods of disease control, including agronomic measures and resistant cultivars, can properly manage crop protection, counteracting pathogen resistance, among others, by reducing the use of synthetic fungicides and applying safe new-generation fungicides based on, e.g., thiosemicarbazone derivatives.

**Author Contributions:** Conceptualization, A.J., B.S.-B., E.M., F.D.; methodology, A.J., E.M., B.S.-B., F.B., F.D., A.G., E.G.; software, A.G., B.S.-B.; validation, A.J.; formal analysis, A.J., E.M., B.S.-B., A.G., F.D.; investigation, A.J.; resources, A.J., A.G., F.D., G.P., F.B. and B.S.-B.; data curation, A.G. and A.J.; writing—original draft preparation, A.J. and B.S.-B.; writing—review and editing, A.J., B.S.-B.; visualization, A.J.; supervision, A.J.; project administration, A.J.; and funding acquisition, A.J., B.S.-B, G.P. All authors have read and agreed to the published version of the manuscript.

**Funding:** This research was funded by the University of Life Sciences in Lublin, Poland (project OKK/DS/1 in 2021), and by the Programme "FIL-Quota Incentivante" of the University of Parma (Italy) co-sponsored by Fondazione Cariparma (Project: "A sustainable approach to curb natural food poisoning by aflatoxins").

**Institutional Review Board Statement:** Not applicable.

**Informed Consent Statement:** Not applicable.

**Data Availability Statement:** The authors declare that the raw data are available from the corresponding author upon reasonable request.

**Conflicts of Interest:** We declare no conflict of interest.

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
