# Peer review of "Effect of Thiosemicarbazone Derivatives and Fusarium culmorum (Wm.G. Sm.) Sacc. Infection of Winter Wheat Seedlings on Their Health Status and Soil Biological Activity"

_agronomy, doi:10.3390/agronomy12010116_

Round 1
Reviewer 1 Report
The paper title "Effect of thiosemicarbazone derivatives and Fusarium culmorum (Wm.G. Sm.) Sacc. infection of winter wheat seedlings on their health status and soil biological activity" by Agnieszka Jamiołkowska et al. submitted to Agronomy reported an effective control of cereal Fusarium diseases through use of thiosemicarbazone derivatives.
Authors measured many plant and soil characteristics in this study. Indeed, it is meaningful and could have good give insights to develop effective disease control strategies.
However, at present stage, I found there are still many improvements that could be done by the authors especially for the manuscript writing and data analyses.
Abstract: This section should be reorganized, since there is no sentence related to background and conclusion. Also, the organization of results needs to be more concise. The abbr. (F20 and F36) were only shown here, while authors used 2actpTS and JTS instead through the whole manuscript.
2. Introduction: please at least introduce the basic information of the selected derivatives. The purpose of conducting this study also needs to be emphasized.
3. Lines 102-113: the product was purchased from Sigma, right? If so, what is the purpose of the downstream analyses (ESI-MS and elemental analyses)? I didn’t see any information of these analyses in the results and indeed it could be removed.
4. What is the drying condition? (Lines 165-166)
5. Lines 168-187: I would suggest authors shorten this part. This is quite classical and authors can cite these references or mention what kind of changes they did.
6. Lines 214-216: There are 31 carbon sources in the eco plate. Authors should mention the detailed information of the five groups or at least cite reference using the groups as well. Can authors indicate how they calculate the microbial activity or AWCD? One of my major concern is that they did not do the correction with blank well, because there are AWCD values of carbon source below zero (Figure 3) if I understand it correctly. If they did not do the correction, I am afraid this is not the data they can use for conclusions. Moreover, I am also confused with the time point of data that authors used for analyses here. They mentioned 120 h (Line 223, 371) and 96 h (Line 334, 351)
7. Lines 220-222: Did authors use One-way ANOVA for comparison? Authors mentioned many times for the factors which influence the detected characteristics. For example, Lines 275-277. However, they didn’t give the summary of anova results on effects of different factors.
8. What is the main idea of figure 4 and 5? What are the two soils? (Line 375) Can the author explain the sentence in Line 380-382 with more details?
9. The authors should be more careful with the use of references. For example, it’s difficult to for me to follow the third reference.
Other comments: The italics of Fusarium could be checked again. Line 98: What is the plant growth meadium? Line 118: 1-L Line 289: Did the author mean 2actpTS/S+Fc and JTS/S+Fc?
Author Response
Please see the attachement.

Reviewer 2 Report
Thank you for submitting of your revision paper.
After careful evaluation of the manuscript title " Effect of thiosemicarbazone derivatives and Fusarium culmorum (Wm.G. Sm.) Sacc. infection of winter wheat seedlings on their health status and soil biological activity", you can find below my comments.
- The work is understandable, correct, and appropriate for the journal.
- The purpose of and problems to solve in the work are clearly stated.
- The purpose or goal of the work is within the journal’s scope.
- The results are interesting and important to researchers in relevant fields of medicinal plants.
- The introductory section adequately explains the framework and problems of the research.
- The figures and tables are easily readable, correct and informative.
- Importance and impact: the presented results are of significant importance and impact to advancement in the relevant field of research. The article is article likely to be cited in the future.
- The manuscript is well written, has important message, and should be of great interest to the readers.
I would like to suggest some corrections:
Some numeric results in the summary (most important quantitative value/s) must be reported.
Please writhe the Fusarium with Italy throughout the manuscript.
At line 106 please write the producer of a Bruker Anova spectrometer, city and country and also for Waters Acquity Ultraperformance ESI-MS spectrometer.
In all tables, SD must also be used for all values and the letter must be written with superscript.
Please standardize all the names of the strains, they are written also with full name and abbreviated.
In table 2 please write the units of measurement in the table not after that.
Best regards
Round 2
Reviewer 1 Report
My questions have been well addressed.